# *Gemella sanguinis* Infective Endocarditis—Challenging Management of an 8-Year-Old with Duchenne Dystrophy and Undiagnosed Congenital Heart Disease: A Case Report

**DOI:** 10.3390/antibiotics12040706

**Published:** 2023-04-04

**Authors:** Cristina Filip, Corina Maria Vasile, Georgiana Nicolae, Irina Margarint, Loredana Popa, Mihaela Bizubac, Gabriela Ganea, Mihaela Rusu, Bruno Murzi, Mihaela Balgradean, Catalin Cirstoveanu

**Affiliations:** 1Department of Pediatrics, “Carol Davila” University of Medicine and Pharmacy, 020021 Bucharest, Romania; cristina.filip@umfcd.ro (C.F.);; 2Department of Pediatric Cardiology, “M.S. Curie” Children’s Clinical Hospital, 041451 Bucharest, Romania; 3Department of Pediatric and Adult Congenital Cardiology, University Hospital of Bordeaux, 33600 Bordeaux, France; 4Ph.D. School Department, “Carol Davila” University of Medicine and Pharmacy, 020021 Bucharest, Romania; 5Department of Pediatric Cardiovascular Surgery, “M.S. Curie” Children’s Clinical Hospital, 041451 Bucharest, Romania; 6Department of Pediatric Nephrology, “M.S. Curie” Children’s Clinical Hospital, 041451 Bucharest, Romania; 7Department of Neonatal Intensive Care, “Carol Davila” University of Medicine and Pharmacy, 020021 Bucharest, Romania; 8Neonatal Intensive Care Unit, “M.S. Curie” Children’s Clinical Hospital, 041451 Bucharest, Romania; 9Fondazione Toscana “Gabriele Monasterio”, 56126 Pisa, Italy

**Keywords:** infective endocarditis, *Gemella sanguinis*, bicuspid aortic valve, Ross operation, antibiotics, congenital heart disease, children

## Abstract

Congenital heart disease (CHD) remains a predisposing cardiac condition for infective endocarditis (IE). Case report: We present the case of 8-year-old boy with no known pre-existing cardiac disease diagnosed with infective endocarditis (IE) with *Gemella sanguinis.* After admission, he underwent transthoracic echocardiography (TTE), which revealed the presence of Shone syndrome with a bicuspid valve, mitral parachute valve and severe aortic coarctation. He developed a paravalvular aortic abscess with severe aortic regurgitation and left ventricle (LV) systolic dysfunction for which he required a complex surgical intervention after six weeks of antibiotic treatment, consisting of Ross operation and coarctectomy, with a complicated postoperative course, cardiac arrest and ECMO support for five days. The evolution was slow and favorable, with no significant residual valvular lesions. However, persistent LV systolic dysfunction and increased muscle enzymes required further investigation to establish a genetic diagnosis of Duchenne disease. As *Gemella* is not considered a frequent pathogen of IE, no current guidelines refer specifically to it. Additionally, the predisposing cardiac condition of our patient is not currently classified as “high-risk” for IE; this is not considered an indication for IE prophylaxis in the current guidelines. Conclusion: This case illustrates the importance of accurate bacteriological diagnosis in infective endocarditis and poses concerns regarding the necessity of IE prophylaxis in “moderate risk” cardiac conditions such as congenital valvular heart disease, especially aortic valve malformations.

## 1. Introduction

Infective endocarditis (IE) represents a rare, life-threatening condition with long-lasting effects, even among patients who successfully survive and recover [1]. IE disproportionately affects people with underlying structural heart disease who develop bacteremia and is increasingly associated with healthcare exposure, particularly in patients with intracardiac prosthetic material. Bacteremia with a pathogenic organism can develop due to complex interactions between invading microorganisms and the host immune system [2]. Once established, IE can affect almost every major organ system in the body.

IE is an inflammatory condition of the valves, endocardium and vascular intima caused by pathogenic species of bacteria and fungi. The underlying lesion usually involves associated valve lesions and vegetation formation. Vegetation detachment can cause arterial embolism, leading to ischemia and necrosis in significant tissues and organs. In recent years, with an aging population and an increase in the number of elderly patients with degenerative valve disease, the application of prosthetic heart valves, implantation of pacemakers and various endovascular treatment technologies have led to an increased incidence of IE [3,4]. Given the reported poor prognosis and high mortality of IE, early diagnosis and intervention are essential and of significant clinical relevance. Clinical diagnostic methods for IE include blood culture and transesophageal echocardiography. Transesophageal echocardiography can indicate and assess the morphology of each cardiac structure, spatial design, activity status and blood flow through the heart. Transthoracic echocardiography, which was specifically developed for detecting vegetation, is the most widely used imaging method for diagnosing IE and can provide an essential basis for clinical diagnosis and treatment [5].

The diagnosis of IE can be challenging as the presentation is usually atypical. Thus, the modified Duke’s criteria establish the diagnosis of IE based on clinical, echocardiographic, microbiological and immunological criteria [6].

*Staphylococcus aureus* is the major microorganism causing IE. Other bacteria—including Viridans group streptococci (VGS), coagulase-negative staphylococci (CoNS) and *Enterococcus* spp.—can also be a cause [7].

The *Gemella* species was discovered in 1960 after being differentiated from the streptococcus group. *Gemella* species are facultatively anaerobic, catalase-negative, Gram-positive cocci that do not form spores and reside in the gastrointestinal tract, genitourinary tract and oral cavity microbiome [8,9]. Due to its frequent misidentification as Viridans group streptococci, *Gemella* is likely a greater cause of clinical disease than is currently recognized. To date, nine species have been recognized by the scientific community, but only seven are associated with human infection. Among these, *G. morbillorum* and *G. haemolysans* are the most common causes of IE; *G. sanguinis*, although rare, mainly affects the heart valves [8,9].

*Gemella sanguinis* is not commonly reported to be involved in the etiopathogenesis of infective endocarditis and therefore is not mentioned in the current IE guidelines [10,11].

According to the 2015 ESC recommendations, the diagnosis of infective endocarditis is based on the assessment of two major criteria or one major criterion and three minor or five minor criteria [10].

The aortic valve is usually the part of the body most frequently affected by *G. sanguinis* endocarditis. Dental caries may not seem related to *G. sanguinis* endocarditis; however, pre-existing valvular heart disease may be an important predisposing factor. Early surgical intervention accompanied by prolonged parenteral antimicrobial therapy is essential for successful treatment [12].

Through this interesting case report, we want to outline the importance of adequate prophylaxis in patients with predisposing congenital heart disease conditions and the necessity of a neonatal cardiovascular screening program among pediatric patients in Romania.

## 2. Case Report

We report the case of an 8-year-old male with no previous medical history initially admitted for moderate upper abdominal pain and abdominal swelling that lasted approximately three weeks. The boy developed a 48 h fever (38 °C) that started four days before presentation. Upon recording a detailed history, we noted that the boy showed intermittent lower limb claudication over long distances (over 500 m) and difficulty climbing stairs for the past 2–3 years. However, these symptoms were ignored because the patient was a highly energetic child who could run and ride a bicycle. At admission, he had no fever but had marked hepatosplenomegaly, pale skin, a grade III/VI holosystolic murmur in the left aortic and interscapular area, absence of femoral pulse, blood pressure (BP) in the right upper arm 145/80 mm Hg, BP in the lower limbs = 90/60 mm Hg, oxygen saturation 98%, muscle hypotrophy in the calf and multiple complicated dental caries. The initial blood tests evidenced Hb = 8.7 g/dL (microchromic hypochromic anemia), mild leukocytosis, positive inflammatory syndrome with C-reactive protein (CRP) 37. 4 mg/dL (normal < 5 mg/dL), erythrocyte sedimentation rate (ESR)= 52 mm/h, negative procalcitonin, elevated liver and muscle enzymes—ALT = 111 IU/L, AST = 88 IU/L, CK = 1377 IU/L, CK-MB = 57 IU/L, NT-proBNP = 850 pg/mL (cut-off < 125 pg/mL).

Admission echocardiography evidenced a suggestive pattern of paravalvular aortic abscess formation with left coronary aortic sinus remodeling, a bicuspid aortic valve with moderate stenosis, minor aortic regurgitation, paravalvular mitral valve with mild mitral stenosis and moderate mitral regurgitation, severe aortic coarctation and hypertrophic left ventricle (LV) with subnormal ejection fraction (EF = 55%).

Our patient met all the modified Duke’s criteria for the diagnosis of IE according to the American Heart Association (AHA) guidelines [10]. The criteria fulfilled by our patient are underlined in Table 1.

Having a history of fever in the context of a patient with a predisposing cardiac condition, an ultrasound suggestive of paravalvular abscess and biological inflammatory syndrome, we considered a high probability of IE and opted for three consecutive blood culture samples at 12 h intervals; after the second one, we started antibiotic treatment with Ceftriaxone at 100 mg/kg/day, Oxacillin at 300 mg/kg/day and Gentamicin at 3 mg/kg/day. At 48 h after incubation, blood cultures were positive with positive cocci of the *Gemella sanguinis* species with good antibiotic susceptibility (Table 2). The organism was identified as *G. sanguinis* by VITEK 2-Compact System (Biomerieux, France). Antimicrobial susceptibility testing was established using the difusimetric method, and the report was based on Clinical and Laboratory Standards Institute (CLSI) recommendations.

After four days of admission, vesper fever (38.2 C) progressively recurred, and the biological inflammatory syndrome slowly developed. Management was changed to Meropenem at 100 mg/kg/day and Linezolid at 25 mg/kg/day (Vancomycin was replaced with Linezolid due to a severe allergic reaction after the second administration), and this was continued for six weeks. In the first week of treatment, the patient started to develop a significant new diastolic murmur, and echocardiography confirmed spontaneous drainage of the perivalvular aortic abscess, resulting in severe aortic insufficiency and dilation of the left aortic coronary sinus (Figure 1a,b), along with a decrease in LVEF (45%).

The evolution was partially favorable throughout the six weeks of treatment, with fever remission until day 7, partially corrected anemia and regression of splenomegaly, but with clinical signs of heart failure. Antibiotic therapy, heart failure, hypertension mediation, Furosemide, Captopril, Amlodipine and dental caries were treated. Inflammatory tests improved gradually, but NT-proBNP levels increased, and muscle enzymes remained elevated, as seen in Table 3.

After completing antibiotic therapy (6 weeks), the patient was in a relatively good clinical condition, with reasonable control of hypertension (secondary to aortic coarctation), dyspnea on moderate exertion, an IV/VI-grade systolic murmur in the aortic area, marked diastolic murmur, mild anemia and regular inflammatory tests. Transthoracic echocardiography (Figure 2) revealed moderate aortic valve stenosis, severe aortic regurgitation, severe aortic coarctation, progressive dilatation and decreased LV systolic function (LVEF = 35%). Unexpectedly, muscle enzymes increased over the next two weeks to a high serum level, especially CK, whose serum level range was 3800–8500 Ui/L. During the first phase of endocarditis evolution, high cardiac, liver and muscle enzyme levels were interpreted in the context of LV systolic dysfunction (secondary to severe acute aortic insufficiency and possible coronary flow steal) associated with a potential adverse reaction of cytolysis-causing drugs. However, the increasing trend of these blood tests, along with some motor features, such as Gowers’ sign and staggering gait, required a neurological examination, raising the suspicion of hereditary muscular dystrophy. Multiplex-ligation-dependent probe amplification (MLPA) was used as the diagnostic test (from a peripheral blood sample) and confirmed the presence of dystrophin mutation and Duchenne disease. Based on the high peri-operative risk of cardiac intervention, the child was referred to a more experienced cardiac surgery unit for surgery. Within four weeks after IE recovery, the boy underwent a complex surgery consisting of a Ross operation (aortic autograft, pulmonary homograft) and coarctectomy. Immediate postoperative evolution was severe, with cardiac arrest being resuscitated 36 h after surgery, requiring five days of extracorporeal membrane oxygenation (ECMO). The evolution was slow and favorable, necessitating challenging discontinuation of mechanical respiratory support and persistence of clinical heart failure and 21 days of hospitalization in the intensive care unit. Initial postoperative echocardiography revealed good neoaortic valve function, good pulmonary homograft functionality, residual large aortic coarctation and severe LV and RV systolic dysfunction (LVEF = 20%). The patient was discharged five weeks after surgery in good clinical condition with improved exercise tolerance.

Echocardiography performed at discharge evidenced slowly improving kinetics in both ventricles, with LVEF = 35% and LVEF = 45% at discharge (Figure 2).

One year after surgery, our patient had a good cardiovascular status with good exercise tolerance and controlled BP on Furosemide and Captopril medication; muscle enzymes were still elevated after surgery (CK = 5156 UI/L, CKMB = 154 UI/L, ALT = 242 UI/L, AST = 147 UI/L, LDH = 649 UI/L), and the athletic deficit was aggravated progressively. The echocardiography showed moderate systolic biventricular dysfunction, with EF = 40%, minor regurgitation of neoaorta, good functionality of pulmonary homograft, mild mitral stenosis and a large residual coarctation (Figure 3).

## 3. Discussion

The *Gemella* species was first reported in 1917 as a Gram-positive, facultative anaerobic coccal microorganism and was initially classified as Neisseriaceae [13,14]. It is challenging for microbiologists to detect *Gemella* spp. isolates. However, these bacteria are discolored during Gram staining and misclassified as Gram-negative organisms. Furthermore, *Gemella haemolysans* can sometimes be misidentified as *Streptococcus viridans* or remain unidentified [15]. Consequently, early accurate diagnosis is crucial for prompt patient management.

Although it is not a very aggressive germ, in this case, it caused a serious evolution of the endocarditis process (even though it presented a good sensitivity to antibiotics) with the eventual prohibition of aortic valvuloplasty due to perivalvular abscess with severe valvular lesions and significant dilatation of the left aortic coronary sinus. The cardiac surgery was very complex but successfully performed in a specialized center. The Ross procedure was the only option; the successful ECMO in a patient with worsening myocardial dysfunction due to Duchenne disease was considered a great success.

Particularly considering our patient’s complex pathological conditions, what is notable is that the young boy was largely asymptomatic until the age of 8 years, with good exercise tolerance (including anamnestic data of good lower limb mobility) despite the presence of Duchenne dystrophy and severe aortic coarctation. The long-term prognosis for the child is primarily related to the progression of Duchenne disease and heart failure due to LV systolic dysfunction in this genetic disease; if the valvular pathology progressively deteriorates with time, ultimately necessitating further cardiac surgery, the peri-operative risk will be high, based on the cardiac and respiratory conditions associated with Duchenne disease. The neurological pathology was unexpected in this case as there is no documented association between congenital heart disease and genetic neuromuscular disease, with very few cases reported in the literature [7].

IE prevention is highly important, considering the high risk of recurrence after a previous episode of acute infective endocarditis and the additional risk of systemic infections secondary to cortisone treatment in Duchenne disease, with the need to maintain proper dental and periodontal health to reduce the risk of bacteremia.

According to the latest ESC and AHA/ACC guidelines regarding endocarditis prophylaxis, our child has a class IIa recommendation for antibiotic treatment periprocedural in case of dental procedures [10], and strict control of infections with risk of bacteremia and reduction in germ carriage is recommended. Based on this case, optimal dental hygiene should be strongly encouraged in the general population, especially considering the existence of undetected cardiac malformations (such as aortic bicuspid and coarctation) and the risk of IE in this context. Our study may suggest that a congenital cardiac malformation could manifest clinically for the first time directly through a complication of the disease—infective endocarditis.

Further discussions may be requested regarding the lack of recommendations for antibiotic prophylaxis of IE for non-high-risk cardiac conditions in current guidelines, as in our case [10,11]. According to the 2015 ESC Guidelines, indications for antibiotic prophylaxis of IE are limited to some specific predisposing cardiac conditions (patients with any prosthetic valve, including a transcatheter valve, or those in whom any prosthetic heart valve repair material has been used; patients with a previous episode of IE or patients with CHD). In our experience, IE frequently occurs in patients with non-high-risk conditions (such as congenital valve malformation), which makes it questionable whether antibiotic prophylaxis is appropriate in patients with intermediate-risk heart disease.

In a recent paper, Rabah et al. reported a similar case in an adult without any predisposing cardiac disease before being diagnosed with *Gemella* endocarditis. After the TTE was performed, it was revealed that the patient had severe mitral regurgitation [16].

Another case of *Gemella* endocarditis was described by Shah et al. [9] in a relatively healthy patient without any known risk factors who developed severe endocarditis caused by *G. sanguinis*, affecting two valves and a paravalvular abscess with subsequent destruction of the aortomitral curtain. The patient ultimately did not survive despite aggressive treatment with broad-spectrum antibiotics, an early surgical intervention and essential postoperative measures, including VV ECMO. This case provides an important illustration of the aggressive nature of *G. sanguinis* and its ability to cause rapid progression of clinical symptoms and pathology.

Additionally, this case illustrates that IE is not always due to typical microorganisms; in recent years, many cases of IE with non-typical germs have been reported. However, the *Gemella* genus is rare and involved in infectious endocarditis, Sanguinis being the least common of all *Gemella* species [8,9,17,18,19]. Less commonly, this germ has been reported to cause IE in pediatric patients, with few cases among children reported in the literature [6,20].

## 4. Conclusions

Our case, the first reported in Romania with *Gemella sanguinis* in a pediatric patient, illustrates that valve surgery, in combination with appropriate antibiotic treatment, could be a successful therapeutic option for infective endocarditis caused by this rarely reported microorganism. We also highlight the importance of infective endocarditis prophylaxis in the presence of identified risk factors.

## 5. Teaching Point

*Gemella sanguinis* is a rare cause of IE, and its diagnosis represents a challenge for clinicians. The primary source of *Gemella* is the oral cavity. Despite its high sensitivity to antibiotics and the implementation of appropriate treatment, the evolution was severe with extensive valve damage, suggesting that *Gemella* species could be very aggressive.

## Figures and Tables

**Figure 1 antibiotics-12-00706-f001:**
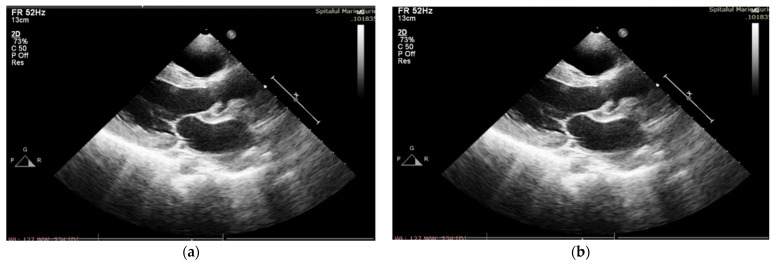
(**a**) TTE PLAX—paravalvular aortic abscess—in the course of formation; (**b**) after spontaneous evacuation.

**Figure 2 antibiotics-12-00706-f002:**
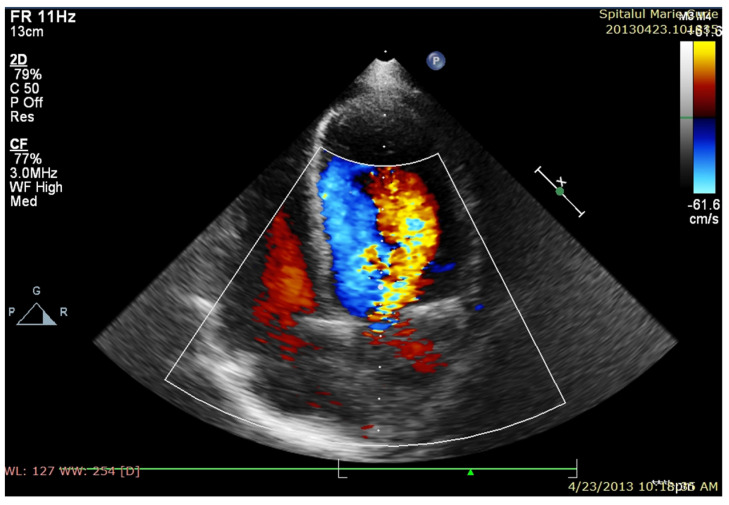
TTE: A5CH (modified): severe aortic regurgitation.

**Figure 3 antibiotics-12-00706-f003:**
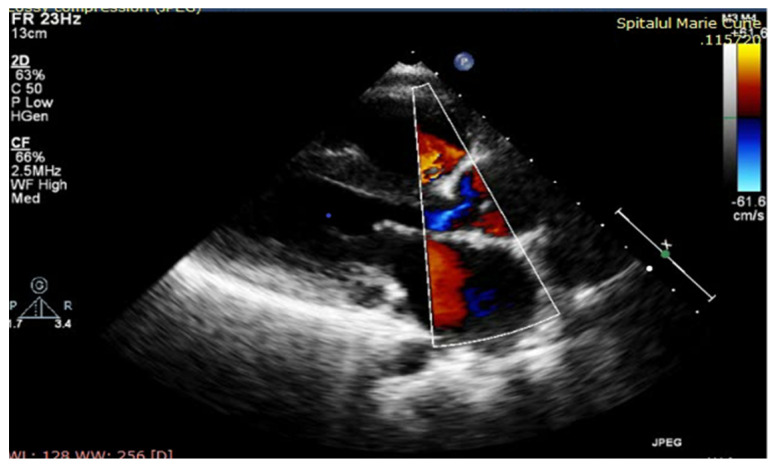
TTE at one-year follow-up: minor regurgitation of neoaorta.

**Table 1 antibiotics-12-00706-t001:** Modified Duke criteria (adapted from Habib G. et al. [10]).

Major Criteria	Minor Criteria
**Blood cultures positive for IE** Typical microorganisms consistent with IE from 2 separate blood cultures: Viridans streptococci, Streptococcus gallolyticus (*Streptococcus bovis*), HACEK group, *Staphylococcus aureus*; Community-acquired enterococci, in the absence of a primary focus. *Microorganisms consistent with IE from persistently positive blood cultures:* *Minimum 2 positive blood cultures of blood samples drawn > 12 h apart*; All of 3 or a majority of ≥4 separate cultures of blood (with last samples drawn ≥ 1 h apart); Single positive blood culture for Coxiella burnetii or phase I IgG antibody titer > 1:800. **Imaging positive for IE** *Echocardiogram positive for IE*: Vegetation; *Abscess*, pseudoaneurysm, intracardiac fistula; Valvular perforation or aneurysm; New partial dehiscence of prosthetic valve. Abnormal activity around the site of prosthetic valve implantation detected via 18F-FDG PET/CT (only if the prosthesis was implanted for >3 months) or radiolabeled leukocytes SPECT/CT. Definite paravalvular lesions according to cardiac CT.	*Predisposing heart condition*, or injection drug use. *Fever*, defined as temperature > 38 °C. Vascular phenomena: major arterial emboli, septic pulmonary infarcts, infectious (mycotic) aneurysm, intracranial hemorrhage, conjunctival hemorrhages and Janeway’s lesions. Immunological phenomena: glomerulonephritis, Osler’s nodes, Roth’s spots and rheumatoid factor. Positive blood culture but does not meet a major criterion as noted above or serological evidence of active infection with organism consistent with IE.

**Table 2 antibiotics-12-00706-t002:** Antibiotic susceptibility of *Gemella sanguinis* in our case.

Antimicrobial Agent	Sensitivity
Penicillin	S
Aminopenicillin	S
Cefazolin	S
Cefotaxime	S
Gentamicin	R
Meropenem	S
Erythromycin	S
Vancomycin	S
Teicoplanin	S
Doxycycline	S
Ciprofloxacin	S

S = sensitive, R = resistant.

**Table 3 antibiotics-12-00706-t003:** Evolution of blood tests throughout the therapy period.

Blood Tests	Initial	After 2 Weeks of Treatment	After 4 Weeks of Treatment	After 6 Weeks of Treatment
Hb(g/dL)	8.7	10.5	10.1	10.2
CRP (mg/L)	37.4	10.2	1	1.1
ALT (UI/L)	111	181	242	216
CK (UI/L)	1377	4967	6816	3648
CK-MB (UI/L)	57	171	266	125
NT-proBNP (pg/mL)	850	1052	894	2311

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
