# Peer review of "Gemella sanguinis Infective Endocarditis—Challenging Management of an 8-Year-Old with Duchenne Dystrophy and Undiagnosed Congenital Heart Disease: A Case Report"

_antibiotics, 2023, doi:10.3390/antibiotics12040706_

Round 1
Reviewer 1 Report
This report describes a case of Gemella sanguinis infective endocarditis (IE) in a boy who was diagnosed both with a severe congenital heart malformation and muscular dystrophia of Duchenne-type. The report is of some interest but it is not well-written. Several claims are made that lack support. I list concerns and comments below.
Concerns and comments
1. Bacteria are denoted by a genus and a species. Genus is always given with a capital letter and species with lower-case letters. Always italics. It is thus “Gemella sanguinis”. Check entire manuscript…
2. Abstract “endocarditis among children appears to have increased in recent years”. This claim lacks support!
3. Abstract “congenital heart disease (CHD), this pathology becoming the predominant underlying condition for IE in children”. Was there ever another more common underlying condition in children? Change!
4. Abstract “seems to be the least involved in endocarditis”. Several similar claims are made throughout the text. I find that the claim is not well supported by the references given. Provide firm support or do not state this.
5. Line 44-46, there is a gap in the logic between bacteremia and IE. Please modify.
6. Line 48, I have never heard of viruses or chlamydia (which is a bacterium by the way) which cause IE. Delete!
7. Line 51, lack of logic since embolism is an infrequent cause of death.
8. Line 63, Duke criteria do not facilitate the diagnosis, they “define” the diagnosis.
9. Line 75-76 “rare, mainly affect the heart valves [8,9].”. The references given do not support the claim.
10. Line 78 what is an “acute IE guideline”??
11. How was species determination of the bacteria performed? This MUST be described.
12. How was antibiotic susceptibility testing performed? Which breakpoints are used? MIC for gentamicin?
13. Line 128-129, allergic reactions to vancomycin are extremely uncommon. Was it not just “red man syndrome” which is not an allergic reaction…
14. Why was vancomycin and linezolid used? The bacterium was sensitive to benzyl-penicillin. Bensyl-penicillin and gentamicin are likely synergetic!
15. Why change to meropenem?? Makes no sense!
16. Line 163, how was genetic testing for Duchenne performed and which were the results. When there are two very rare conditions found at the same time, one should always consider that they (the heart malformation and muscle disease) both stem from a common genetical defect…
17. Line 192 “an on-going” disease. Not a proper way to put the claim. Rephrase.
18. Line 194, the sentence starts in the middle of a thought. Rewrite.
19. Line 199 “Therefore, accurately diagnosing such an organism is crucial to promptly managing patients.”. To treat a patient with IE you do not need to know the species of the bacterium. This will hardly affect the management.
20. Line 205 “very good surgeon” is a lame claim. Delete! Specialized center is better.
21. Line 205-207, I do not understand what is meant here!
22. Line 216-218 frames the entire core of this case. This could be expanded.
23. Line 219 and onwards on prophylaxis. It is unclear what type of prophylaxis that is referred to. For adults there are discussions about prophylaxis during dental extractions, but is some form of continuous antibiotic treatment referred to here? Please explain!
24. 235-237, when should such persons receive prophylaxis and what type of prophylaxis is meant? Please rewrite!
25. Line 249, “Also, this case illustrates the extensive broad of bacteria”. This claim is by default not correct. A single case with a single bacterium cannot illustrate that IE can be caused by many different bacteria. Delete!
26. Line 251-252, “endocarditis, Sanguinis being the rarest of all Gemella species [8,9,18,19]”. What is a “rare species”? I suspect that what you mean is that G. sanguinis is the least common of Gemella species isolated in IE. To support such a claim, case reports are not appropriate. Use case series instead. For example: https://academic.oup.com/ofid/article/6/10/ofz437/5580803?login=false
27. Line 260, “The primary source of Gemella is the oral cavity” might be a relevant teaching point but it is not mentioned in the article before it comes as a teaching point!
28. Line 260-261 “Early recognition of this pathogen results in prompt treatment and avoidance of complications.”. This claim is lame, the pathogen does not need to be recognized to be treated. Essentially any antibiotic would be effective in this case and early recognition of bacterial etiology is not a prerequisite for treatment.
29. Line 262-263 “We want to highlight the importance of cardiac screening in the pediatric population 262 in the early childhood years.”. This is not really a good teaching point from this case. The heart condition of this boy could have been detected earlier through screening with stetoscopy but is this really a “teaching point”?
Author Response
Dear reviewer,
We highly appreciate your comments.
Concerns and comments
- Bacteria are denoted by a genus and a species. Genus is always given with a capital letter and species with lower-case letters. Always italics. It is thus “Gemella sanguinis”. Check entire manuscript…
We modified the errors.
- Abstract “endocarditis among children appears to have increased in recent years”. This claim lacks support!
We have improved our abstract presentation.
- Abstract “congenital heart disease (CHD), this pathology becoming the predominant underlying condition for IE in children”. Was there ever another more common underlying condition in children? Change!
We have improved our abstract presentation.
- Abstract “seems to be the least involved in endocarditis”. Several similar claims are made throughout the text. I find that the claim is not well supported by the references given. Provide firm support or do not state this.
We have improved our references to support the statements.
- Line 44-46, there is a gap in the logic between bacteremia and IE. Please modify.
We modified it accordingly.
- Line 48, I have never heard of viruses or chlamydia (which is a bacterium by the way) which cause IE. Delete!
We modified it accordingly.
- Line 51, lack of logic since embolism is an infrequent cause of death.
We modified it accordingly.
- Line 63, Duke criteria do not facilitate the diagnosis, they “define” the diagnosis.
We modified it accordingly.
- Line 75-76 “rare, mainly affect the heart valves [8,9].”. The references given do not support the claim.
We modified it accordingly.
- Line 78 what is an “acute IE guideline”??
We modified it accordingly.
- How was species determination of the bacteria performed? This MUST be described.
- the first 2 blood cultures proceeded in our hospital lab, where the Gemella sanguinis species was identified by VITEK 2-Compact System (Biomerieux, France) on Bactec cultures but the antibiotic susceptibility was not tested, taking into consideration that it isn’t standardized. Based on infectious diseases literature (Koneman’s Gram-Positive Coci Part II: Streptococci, Enterococci and the “Streptococcus-Like Bacteria” in Color Atlas and Textbook of Diagnosis Microbiology, Sixth Edition,2006), was considered that Gemella has a sensibility to Penicillin, Ampicillin, Rifampicin, and Vancomycin; there are some Gemella species with a low resistance to Aminoglycosides and Trimetoprim. We sent another blood sample to a private out-hospital lab where was performed antibiotic susceptibility based on CLSI guidelines.
- How was antibiotic susceptibility testing performed? Which breakpoints are used? MIC for gentamicin?
- Unfortunately, we did not get a response from the central lab, where the tests were performed, but we didn’t get MIC for gentamicin.
- Line 128-129, allergic reactions to vancomycin are extremely uncommon. Was it not just “red man syndrome” which is not an allergic reaction…
There was a full-body skin eruption and angioedema during the second dose of Vancomycin (i.v. infusion over 60 minutes, 15 mg/kg/dose, dilute solution 5mg/ml); it could not be distinguished from red man syndrome, but the symptoms were remitted after corticoids administration. For safety reasons, it was decided to switch vancomycin with Linesolid.
- Why was vancomycin and linezolid used? The bacterium was sensitive to benzyl-penicillin. Bensyl-penicillin and gentamicin are likely synergetic!
The infectious disease specialist established the treatment; because after 4 days of Ceftriaxone +Oxacylin+Gentamicin, the fever relapsed, the inflammatory tests increased again (after an initial decrease), and the valve damage was progressively very severe. It was decided to start with Meropenem with Vancomycin.
- Why change to meropenem?? Makes no sense!
We revised the antibiogram results.
- Line 163, how was genetic testing for Duchenne performed and which were the results. When there are two very rare conditions found at the same time, one should always consider that they (the heart malformation and muscle disease) both stem from a common genetical defect…
Multiplex ligation-dependent probe amplification (MLPA) has been used as the diagnostic test (from a peripheral blood sample) and confirmed the presence of dystrophin mutation and Duchenne disease.
- Line 192 “an on-going” disease. Not a proper way to put the claim. Rephrase.
We have revised it.
- Line 194, the sentence starts in the middle of a thought. Rewrite.
We have revised it.
- Line 199 “Therefore, accurately diagnosing such an organism is crucial to promptly managing patients.”. To treat a patient with IE you do not need to know the species of the bacterium. This will hardly affect the management.
- Line 205 “very good surgeon” is a lame claim. Delete! Specialized center is better.
We revised.
- Line 205-207, I do not understand what is meant here!
We revised.
- Line 216-218 frames the entire core of this case. This could be expanded.
We modified it accordingly.
- Line 219 and onwards on prophylaxis. It is unclear what type of prophylaxis that is referred to. For adults there are discussions about prophylaxis during dental extractions, but is some form of continuous antibiotic treatment referred to here? Please explain!
- 235-237, when should such persons receive prophylaxis and what type of prophylaxis is meant? Please rewrite!
- Line 249, “Also, this case illustrates the extensive broad of bacteria”. This claim is by default not correct. A single case with a single bacterium cannot illustrate that IE can be caused by many different bacteria. Delete!
We revised.
- Line 251-252, “endocarditis, Sanguinis being the rarest of all Gemella species [8,9,18,19]”. What is a “rare species”? I suspect that what you mean is that G. sanguinis is the least common of Gemella species isolated in IE. To support such a claim, case reports are not appropriate. Use case series instead. For example: https://academic.oup.com/ofid/article/6/10/ofz437/5580803?login=false
We revised.
- Line 260, “The primary source of Gemella is the oral cavity” might be a relevant teaching point but it is not mentioned in the article before it comes as a teaching point!
We revised.
- Line 260-261 “Early recognition of this pathogen results in prompt treatment and avoidance of complications.”. This claim is lame, the pathogen does not need to be recognized to be treated. Essentially any antibiotic would be effective in this case and early recognition of bacterial etiology is not a prerequisite for treatment.
We revised.
- Line 262-263 “We want to highlight the importance of cardiac screening in the pediatric population 262 in the early childhood years.”. This is not really a good teaching point from this case. The heart condition of this boy could have been detected earlier through screening with stetoscopy but is this really a “teaching point”?
We revised.
Reviewer 2 Report
Thanks for allowing me to review this exciting case study submitted by Filip et al. the manuscript looks fine, with factual findings. I've no significant critics of the authors as they put much effort into solidifying their findings with a well-designed and written case report. My only recommendation is to do a language proofreading through a native speaker to enhance the content better.
Author Response
Dear Reviewer,
We highly appreciate your kind words regarding our work. A native speaker performed an English revision.
Best regards,
Reviewer 3 Report
Please revise the keywords.
Please improve the introduction.
Please mention objectives at thee end of introduction.
Please mention the reason or remove the highlighted text in Table 1 .
Please explain the abbreviation (S and R) in table 2 description.
Please improve the conclusion with experimental finding, too short in this current form.
Please recheck the grammar issues.
Author Response
Dear Reviewer,
We have revised the keyword and improved our introduction.
We added the objectives of our study at the end of the introduction.
The explanation for the underlined text is already in the manuscript.
The criteria met by our patient are underlined (Table 1).
We added the explanation for the abbreviations: S= sensitive and R =resistant.
We improved our conclusion.
An English revision was performed.
Best regards,
Round 2
Reviewer 1 Report
The revision has improved the manuscript satisfactorily.